

# Identification of bacteria on Thai banknotes and coins using MALDI-TOF mass spectrometry and their phenotypic antimicrobial susceptibility profiles

Nattamon Niyomdecha[1], Suwitchaya Sungvaraporn[1], Arisa Pinmuang[1], Narissara Mungkornkaew[2], Thanchira Saita[3], Waratchaya Rodraksa[3], Achiraya Phanitmas[3], Nattapong Yamasamit[3] and Pirom Noisumdaeng[3,4]

[1] Department of Medical Technology, Faculty of Allied Health Sciences, Thammasat University, Khlong Luang, Pathum Thani, Thailand
[2] Microbiology Laboratory, Thammasat University Hospital, Thammasat University, Khlong Luang, Pathum Thani, Thailand
[3] Faculty of Public Health, Thammasat University, Khlong Luang, Pathum Thani, Thailand
[4] Thammasat University Research Unit in Modern Microbiology and Public Health Genomics, Thammasat University, Khlong Luang, Pathum Thani, Thailand

Corresponding author
Pirom Noisumdaeng,
pirom.n@fph.tu.ac.th

## ABSTRACT

**Background.** The existence and transmission of pathogenic and antibiotic-resistant bacteria through currency banknotes and coins poses a global public health risk. Banknotes and coins are handled by people in everyday life and have been identified as a universal medium for potentially microbial contamination.

**Methods.** To ascertain existence of medically important bacteria, a total of 300 samples including 150 banknotes and 150 coins were randomly collected at onsite retail fresh meat stores, *i.e.*, pork and chicken, fish, and seafood stores, from nineteen fresh markets distributed across Bangkok, Thailand. An individual banknote or coin was entirely swabbed, and bacterial culture was carried out using tryptic soy agar (TSA), sheep blood agar (SBA) and MacConkey agar (Mac). A colony count was performed and bacterial species identification was conducted using matrix-assisted laser desorption/ionization (MALDI)-time of flight (TOF) mass spectrometry. Phenotypic antimicrobial susceptibility testing was carried out using the Kirby–Bauer disc diffusion methods.

**Results.** The results demonstrated that the bacterial contamination rate was higher on banknotes than on coins (93.33% *vs.* 30.00%) in all three store types. A substantial number of colonies of >3,000 colony forming units (CFU) was predominantly found in banknotes (70.00%), especially from fish store (83.3%); meanwhile, <1,000 CFU was observed in coin sample (76.67%). MALDI-TOF mass spectrometry could identify 107 bacterial species, most of them were *Staphylococcus kloosii* (14.02%, 15/107), *Staphylococcus saprophyticus* (12.15%, 13/107), and *Macrococcus caseolyticus* (8.41%, 9/107). The prevalence based on genera were *Staphylococcus* (36.45%, 39/107), *Acinetobacter* (20.56%, 22/107), and *Macrococcus* (10.28%, 11/107). Almost all Staphylococcus isolates had low susceptibility to penicillin (21%). Notably, *Staphylococcus arlettae*, *Staphylococcus haemolyticus* and *M. caseolyticus* were multidrug-resistant (MDR). It is notable that none of the staphylococci and macrococci isolates exhibited inducible clindamycin resistance (D-test negative). *Escherichia coli* and *Pseudomonas putida*

isolates were carbapenem-resistant, and *Acinetobacter baumannii* isolates were MDR with showing carbapenem resistance.

**Conclusion**. Our data demonstrated a high prevalence of medically important bacteria presented on Thai currency, which may pose a potential risk to human health and food safety. Food vendors and consumers should be educated about the possible cross-contamination of bacteria between the environment, food item, and currency.

# INTRODUCTION

Cashless payment technology is a feature that currency transactions in the digital world provide for individuals; nevertheless, this advanced technology is still unable to apply to all users or be implemented in all places. Consequently, transactions utilizing banknotes and coins remain significant and unavoidable. Accumulated data reported over the last 20 years globally on the microbial status and survival of pathogen on currency notes indicated that this could represent a potential cause of disease transmission, especially respiratory and gastrointestinal infections (*Schaarschmidt, 1884*; *Agersew, 2014*; *Ofoedu et al., 2021*). The frequent handling of money by individuals with bacteria-contaminated hands, as well as its contact with contaminated surfaces or food, poses a significant risk. Money, as a universal medium of exchange, has the potential to act as a carrier for the transmission of infectious pathogens to humans and animals. Additionally, it may facilitate the horizontal transfer of antimicrobial resistance genes between commensal and pathogenic bacteria, as documented in prior studies (*Nemeghaire et al., 2014*; *Chanchaithong, Perreten & Schwendener, 2019*).

Previous reports indicated that banknotes and coins were frequently contaminated with bacteria, particularly from human skin microbiota such as coagulase-negative staphylococci (CoNS), gut microbiota enterococci, and environmental *Bacillus* spp. (*Ofoedu et al., 2021*; *Meister et al., 2023*). Additionally, various other pathogenic bacteria, such as *Staphylococcus aureus* (*S. aureus*), *Streptococcus* spp., *Escherichia coli* (*E. coli*), *Proteus* spp., *Klebsiella* spp., *Enterobacter* spp., *Pseudomonas* spp., *Acinetobacter* spp., *Salmonella* spp., and multiple-drug resistance bacteria, are frequently identified (*Meister et al., 2023*). Certain bacteria have been documented to flourish on the skin (*Mackintosh & Hoffman, 1984*), and on inanimate surfaces for a long time (*Kramer, Schwebke & Kampf, 2006*). Nonetheless, several parameters such as humidity, temperature, pH, salinity, surface material, UV radiation, the presence of organic matter, and ventilation, along with pathogen-specific factors like the initial quantity of infectious agents, can significantly influence the persistence of infectious microbes (*Leung, 2021*).

Thailand is located in a tropical region with high temperatures and humidity, which effectively encourages the growth of germs. Thailand is experiencing rapid aging, positioning itself as the second most aged society in ASEAN, following Singapore. At present, twenty percent of the Thai population is aged 60 years or older. By 2030,

approximately one-third of the Thai population is projected to be over 60 years old (*UNFPA Country Office in Thailand, 2021*). These factors might raise the risk of bacterial colonization, transmission, and infection in elderly individuals who are susceptible. Consequently, it is imperative to prioritize the evaluation of bacterial contamination on currency to reduce public health hazards, particularly for vulnerable populations that lack access to advanced cashless technologies. Unfortunately, research is scarce on this significant field in Thailand. This study aimed to identify bacterial strains contaminating Thai currency using the MALDI-TOF assay and to determine the antimicrobial resistance profiles of the medically important bacteria detected. This has the potential to offer important information regarding public health surveillance, which could encourage individuals to be aware of and maintain their hygiene.

## MATERIALS AND METHODS

### Study sites and sample collection

This cross-sectional study was conducted from June to July 2024, which any kinds of Thai banknotes and coins were randomly collected from retail fresh meat stores at the markets in Bangkok, Thailand. Excellent fresh food markets certified by the Department of Internal Trade (DIT), Ministry of Commerce, Thailand, located in Bangkok were selected as the study sites.

Based on previous investigations, many studies from several countries including Thailand have reported the very high prevalence of bacterial contamination ranging from 69% to 100% of tested currencies with varying degrees of sample sizes randomly collected (70–343 banknotes and coins) (*Phunpae et al., 2018*; *Ofoedu et al., 2021*; *Yar, 2020*; *Ejaz, Javeed & Zubair, 2018*; *Gabriel, Coffey & O'Mahony, 2013*; *Alemu, 2014*). In addition, most of the tested currencies were contaminated by pathogenic or potentially pathogenic bacteria. Therefore, we hypothesized that most of actively used banknotes and coins in the circulation system might be contaminated by bacteria. Our study set the sample sizes with at least 35 samples based on each type of fresh stores (pork and chicken, seafood, and fish stores) located in nineteen markets, identified as excellent fresh food markets, across Bangkok. Therefore, a presumptive sample sizes of 105 banknotes and 105 coins should be collected. However, to precise statistically descriptive analysis, our study collected a total of 300 samples, including 150 banknotes and 150 coins, with a similar proportion from each store type as described in Table 1. The samples were collected by the purchasing process from three categories of retail stores: pork and chicken stores, fish stores, and seafood stores. The samples received from each store were kept in a sterile plastic bag under cold storage condition before further processing.

### Sample processing

Each banknote or coin was entirely swabbed, and the swab was then resuspended with 1 ml of tryptic soy broth (TSB) and incubated for 30 min at 37 °C (*Phunpae et al., 2018*). One-hundred microliters ($\mu$l) TSB culture was directly inoculated to tryptic soy agar (TSA) without making a dilution for colony count, and a full loop of culture was utilized to isolate the bacterial colonies in the sample using a cross-streak technique on sheep blood agar

Table 1 Details of markets and data distribution in sample collection.

| Market details[a] | | Pork/Chicken | | Seafood | | Fish | | Total | |
| --- | --- | --- | --- | --- | --- | --- | --- | --- | --- |
| Label | District | B | C | B | C | B | C | B | C |
| (I) Phra Nakhon Zone | | | | | | | | | |
| A | Sai Mai | 3 | 3 | 3 | 3 | 1 | 3 | 7 | 9 |
| B | Bang Sue | 3 | 3 | 3 | 3 | 3 | 3 | 9 | 9 |
| C | Bang Khen | 3 | 2 | 3 | 3 | 3 | 3 | 9 | 8 |
| D | Bang Khen | 3 | 3 | 3 | 3 | – | 3 | 6 | 9 |
| E | Khan Na Yao | 3 | 3 | 3 | 3 | 3 | 2 | 9 | 8 |
| F | Min Buri | 2 | 3 | 1 | 3 | 1 | 3 | 4 | 9 |
| G | Lak Si | 3 | 4 | 4 | 4 | 3 | 4 | 10 | 12 |
| H | Chatuchak | 2 | 2 | 3 | 2 | – | – | 5 | 4 |
| I | Suan Luang | 3 | 3 | 3 | – | 2 | – | 8 | 3 |
| J | Prawet | 2 | 2 | 3 | 2 | 2 | 2 | 7 | 6 |
| K | Din Daeng | 4 | 3 | 4 | 3 | 4 | 4 | 12 | 10 |
| L | Pathum Wan | 4 | 4 | 4 | 4 | – | – | 8 | 8 |
| (II) Thonburi Zone | | | | | | | | | |
| M | Bangkok Noi | 4 | 3 | 4 | – | 1 | – | 9 | 3 |
| N | Khlong San | 3 | 4 | 4 | 4 | 1 | – | 8 | 8 |
| O | Taling Chan | 6 | 7 | 5 | 5 | – | – | 11 | 12 |
| P | Suan Luang | 4 | 4 | 4 | 4 | – | 4 | 8 | 12 |
| Q | Thung Khru | 3 | 3 | 2 | 5 | 2 | 2 | 7 | 10 |
| R | Thawi Watthana | 1 | 3 | 3 | – | 1 | 1 | 5 | 4 |
| S | Thawi Watthana | 3 | – | 2 | 3 | 3 | 3 | 8 | 6 |
| | Total | 59 | 59 | 61 | 54 | 30 | 37 | 150 | 150 |

Notes.
[a] Samples were collected from each market once, except for market O, which was sampled twice.

Abbreviations: B, banknote; C, coin.

(SBA) and MacConkey agar (Mac). All culture plates were incubated overnight at 37 °C. The leftover TSB culture was kept at 4 °C. A colony count on TSA was reported as CFU/ml or too many to count (TMTC). The positive culture on SBA/Mac was recorded and calculated as the prevalence percentage (%). The suspected medically important bacterial colonies were chosen for subculturing as pure isolation on SBA/Mac, and the basic characteristics was determined by gram-staining to differentiate between gram-positive and gram-negative bacteria and by testing the presumptive tests of catalase/oxidase (*Lee, 2021*).

## Colony identification using MALDI-TOF mass spectrometry

A fresh pure colony was spotted on a MALDI-TOF target Microflex (Bruker Daltonik, Wissembourg, France) using a sterile wooden toothpick. An expanded direct transfer (''on-target'' extraction) method was used for the gram-positive colony. The spotted colony as a film was treated with one µl of 70% formic acid (Sigma Aldrich, St. Louis, MO, USA) in HPLC grade water (Sigma Aldrich, St. Louis, MO, USA) and left to air dry at room temperature. Following the drying, 1–2 µl of a saturated solution of α-Cyano-4-hydroxycinnamic acid (HCCA) matrix solution in 50% acetonitrile and 2.5% trifluoroacetic acid was applied and allowed to dry at room temperature for 5 min before

loading sample targets into the matrix-assisted laser desorption/ionization (MALDI)-time of flight (TOF) mass spectrometry (MALDI-TOF MS) instrument for analysis. On the other hand, a direct transfer method was applied for the gram-negative colony. The same protocol was done as described for the gram-positive colony except for not treating with formic acid solution (*Calderaro & Chezzi, 2024*).

Sample mass spectra were acquired and analyzed by the Biotyper software to compare the protein profile of the bacteria with a library database (*Calderaro & Chezzi, 2024*). An analyzed score <1.70 was considered unreliable and no identification of bacteria, while a score ≥1.70 was accepted for identification. A score >1.99 represents a precise identification of genus and species, whereas a score ≥1.70−1.99 confirms identification of genus only, without species specification (*Calderaro & Chezzi, 2024*; *Czeszewska-Rosiak et al., 2025*).

### Antimicrobial susceptibility test

An antimicrobial susceptibility assay was performed using the Kirby-Bauer disc diffusion method. Inhibition zones were measured and interpreted using the Clinical and Laboratory Standards Institute (CLSI) guidelines (33rd edition) (*Institute CLSI, 2023*). The following antibiotics (Oxoid) were used: penicillin 10 µg (P), ampicillin 10 µg (AMP), ampicillin/clavulanic acid 20/10 µg (AMC), amikacin 30 µg (AK), clindamycin two µg (DA), erythromycin 15 µg (E), cefoxitin 30 µg (FOX), sulfamethoxazole/trimethoprim 23.75/1.25 µg (SXT), tetracyclin 30 µg (TE), fusidic acid 10 µg (FD), gentamicin 10 µg (CN), gentamicin 120 µg (CN), vancomycin 30 µg (VA), ceftazidime 30 µg (CAZ), cefotaxime 30 µg (CTX), meropenem 10 µg (MEM), imipenem 10 µg (IMP), ciprofloxacin five µg (CIP), and levofloxacin five µg (LEV). The clinical resistance breakpoints established for *Staphylococcus* spp. were applied to *Macrococcus* spp., as proposed by *Cotting et al. (2017)*. Inducible clindamycin resistance was detected using a double disk diffusion test by CLSI guidelines. The antimicrobial susceptibility results for each group of tested microorganisms were calculated as a percentage of susceptibility (only S result) and presented as antimicrobial resistance profiles.

### Statistical analysis

Descriptive statistical analysis was carried out to calculate the frequency or percentage of data.

### Biosafety approval

The biosafety issue was approved by the Institutional Biosafety Committee (TU-IBC), Thammasat University with the certificate of approval number 015/2567.

## RESULTS

### Sample collection and bacteria identification by MALDI-TOF mass spectrometry

Nineteen excellent fresh food markets (labelled A-S in Fig. 1) located in different locations across two zone areas of Thonburi (green zone) and Phra Nakhon (yellow zone) of Bangkok province were chosen as study sites. A total of 300 samples, including 150 banknotes and

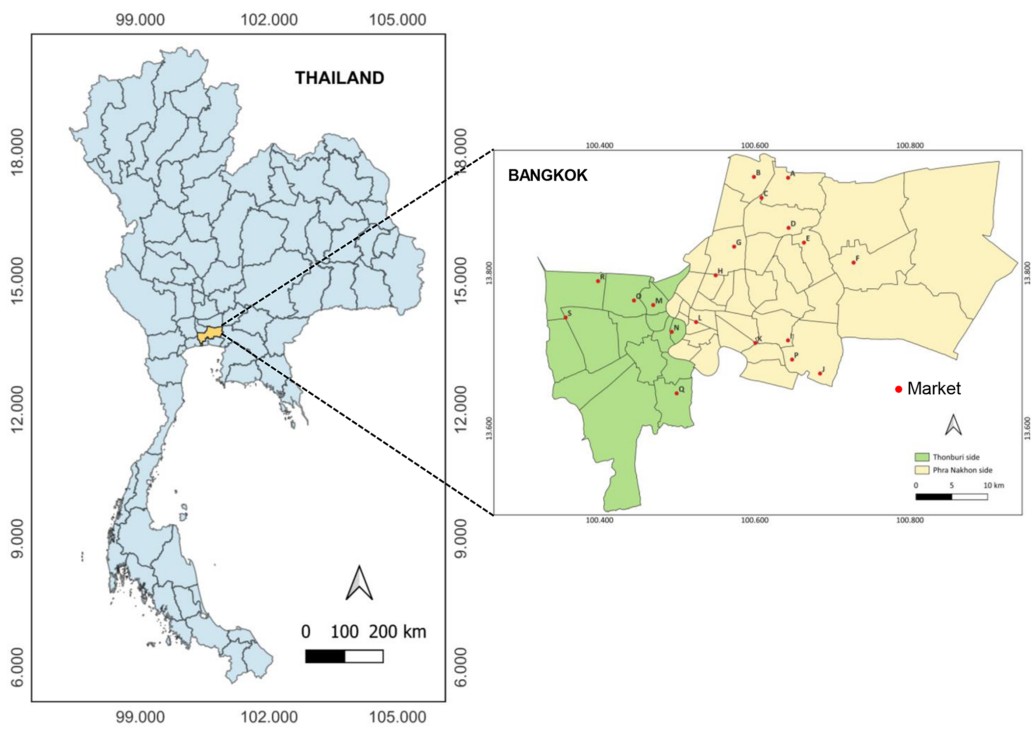

**Figure 1** Map of the study sites encompassing nineteen excellent fresh food markets (labeled A-S) located in different locations across two zone areas of Thonburi (green zone) and PhraNakhon (yellow zone) of Bangkok, Thailand.

150 coins, were collected with a similar proportion from each store type as described in Table 1.

As shown in Fig. 2, the bacteria contamination rates were detected on banknotes higher than on coins (93.33% *vs.* 30.00%) in all store types. All positive samples were contaminated with multiple colony types. The highest bacterial contamination rate on banknotes was observed in fish stores (100%, 30/30), whereas the highest contamination rate on coins was found in pork and chicken stores (45.76%, 27/59) (Table S1). Additionally, the colony count was evaluated for banknote and coin samples as illustrated in Figs. 3A and 3B. Seventy percent of the banknote samples exhibited a substantial number of colonies, ranging from over 3,000 CFU to TMTC, whereas 76.67% of the coin samples displayed colony counts below 1000 CFU. Notably, the highest percentage of >3000-TMTC CFU colony count was observed on banknotes at fish stores (83.33%) and on coins at pork and poultry stores (13.56%) (Table S2). In addition, the number of contaminated bacteria detected across different location sites for both banknote and coin was comparable by showing minimum detection at ranged of 0–60 CFU and maximum detection of >3,000 CFU or TMTC (Table S2).

One-hundred and twenty-five colonies grown on SBA/Mac were randomly selected from any kind and any source of samples. They were isolated and identified using the MALDI-TOF MS, and the analyzed score of each isolate was reported in Table S3. About

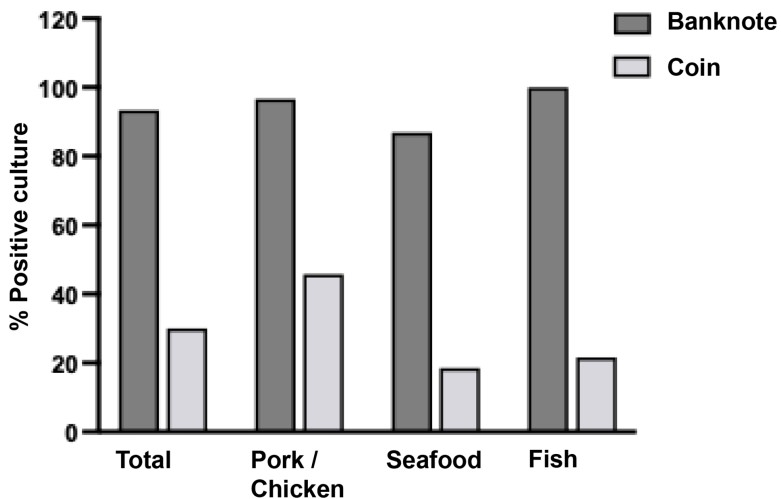

**Figure 2** Percentage of bacterial detection on Thai banknotes and coins from retail fresh meat stores, *i.e.,* pork and chicken, fish, and seafood stores.

85.60% (107/125) of isolates could be reported due to some of them showing the problems of no peak signal presence or unidentified peptide peak based on the database. Colony isolation from fish stores could be identified in the highest proportion (90.91%, 20/22) when compared to pork and chicken stores (89.66%, 52/58), and seafood stores (77.78%, 35/45). Data from MALDI-TOF MS indicated that the majority were uncommon bacterial strains identified as shown in Table 2. Of the detected strains, 58.69% (27/46) were gram-negative. The top three predominant strains were *Staphylococcus kloosii* (14.02%, 15/107), *Staphylococcus saprophyticus* (12.15%, 13/107), and *Macrococcus caseolyticus* (8.41%, 9/107). *Staphylococcus kloosii* and *Macrococcus caseolyticus* were the most prevalence in pork and chicken stores; meanwhile, *S. saprophyticus* was mostly detected in seafood stores. Nevertheless, the prevalence data analyzed based on genus, *Staphylococcus* (36.45%, 39/107), *Acinetobacter* (20.56%, 22/107), and *Macrococcus* (10.28%, 11/107) were identified as the three most predominant genera.

Considering the majority of uncommon bacterial strains were detected, their category as human pathogens and their pathogenesis was examined, as illustrated in Table 2. The literature review identified eight classifications of infection systems for each microorganism: respiratory tract (RT), skin and soft tissues (SST) or wound, urinary tract (UT), central nervous system (CNS), blood (BL), gastrointestinal tract (GI), other systems, and no report. Among them, the majority of detected bacteria (78.26%, 36/46) were classified as human pathogens, with a significant proportion (69.44%, 25/36) responsible for multiple tract infections. Nonetheless, bloodstream infection was the most prevalent (Fig. 4 and Table S4).

## Antimicrobial susceptibility

As shown in Table 3, fifty bacterial isolates were randomly selected from groups of bacteria that can be tested by standard antimicrobial drugs, which have interpretation criteria based

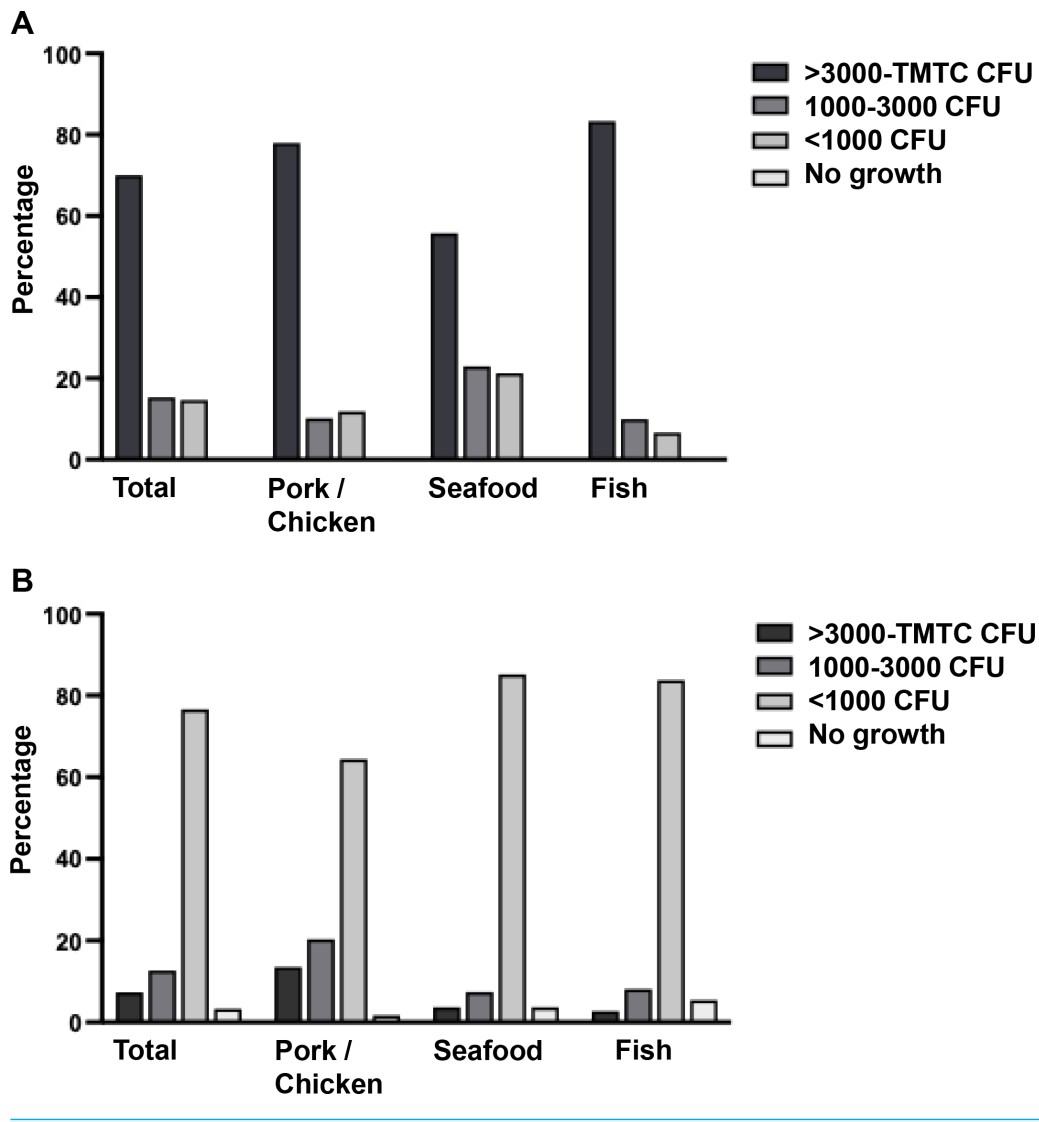

**Figure 3 Enumeration of bacterial colonies grown in tryptic soy agar (TSA).** A colony count was evaluated and reported as CFU/ml or too many to count (TMTC) for banknotes (A) and coins (B).

on CLSI guidelines. Almost all *Staphylococcus* isolates had low susceptibility to penicillin (21%), with the exception of *S. epidermidis*, which demonstrated no resistance for all drugs tested. Isolates of *S. arlettae* and *S. haemolyticus* were multidrug-resistant (MDR), defined as the bacterial isolate acquired non-susceptibility to at least one agent in three or more antimicrobial categories (*Magiorakos et al., 2012*), and methicillin-resistant. Macrococci, a close relative of the *Staphylococcus* genus, were highly resistant to erythromycin and clindamycin. *M. caseolyticus* exhibited intermediate susceptibility to fusidic acid and low susceptibility to tetracyclin. Thus, some *M. caseolyticus* isolates were multidrug resistance. It is notable that none of the staphylococci and macrococci isolates exhibited inducible clindamycin resistance (D-test negative). A single isolate of *Enterococcus* was susceptible to all drugs that were tested.

**Table 2  Description of bacteria identified by MALDI-TOF mass spectrometry, sources of isolation, and infection system.**

| Microorganisms | Gram | No. of isolation | | | Total (%) (N = 107) | Infection systems | | | | | | | | Reference |
|---|---|---|---|---|---|---|---|---|---|---|---|---|---|---|
| | | Pork/Chicken (N = 52) | Seafood (N = 35) | Fish (N = 20) | | RT | SST | UT | CNS | BL | GI | Others | No report | |
| *Acinetobacter baumannii* | GNC/GNCB | 2 | 1 | 0 | 3 (2.80) | x | x | x | x | x | | | | *Howard et al. (2012), Towner (2009)* |
| *Acinetobacter bereziniae* | GNCB/GNB | 0 | 4 | 0 | 4 (3.74) | | | | | x | | | | *Dahal, Paul & Gupta (2023)* |
| *Acinetobacter defluvii* | GNCB/GNB | 1 | 0 | 0 | 1 (0.93) | | | | | | | x | | – |
| *Acinetobacter gandensis* | GNCB/GNB | 1 | 0 | 0 | 1 (0.93) | | | | | | | x | | – |
| *Acinetobacter indicus* | GNCB/GNB | 0 | 0 | 1 | 1 (0.93) | | | | | | | x | | *Dahal, Paul & Gupta (2023)* |
| *Acinetobacter junii* | GNCB/GNB | 1 | 0 | 0 | 1 (0.93) | | | x | | | | | | *Abo-Zed, Yassin & Phan (2020)* |
| *Acinetobacter lactucae* | GNCB/GNB | 1 | 0 | 0 | 1 (0.93) | | | | | | | x | | *Sheck et al. (2023)* |
| *Acinetobacter pittii* | GNCB/GNB | 0 | 1 | 0 | 1 (0.93) | x | x | | | x | | | | *Bello-López et al. (2024)* |
| *Acinetobacter radioresistens* | GNCB/GNB | 0 | 0 | 1 | 1 (0.93) | x | | | | | | | | *Wang et al. (2019)* |
| *Acinetobacter seifertii* | GNCB/GNB | 0 | 1 | 0 | 1 (0.93) | | | | | | | x | | *Sheck et al. (2023)* |
| *Acinetobacter ursingii* | GNCB/GNB | 5 | 0 | 1 | 6 (5.61) | | | | | x | | | | *Dahal, Paul & Gupta (2023)* |
| *Acinetobacter variabilis* | GNCB/GNB | 0 | 1 | 0 | 1 (0.93) | | | | | | | x | | – |
| *Aerococcus viridans* | GPC | 0 | 2 | 1 | 3 (2.80) | | x | | | x | | x | | *Mohan et al. (2017)* |
| *Aeromonas veronii* | GNB | 1 | 0 | 0 | 1 (0.93) | x | x | x | | x | x | | | *Janda & Abbott (2010)* |
| *Bacillus pumilus* | GPB | 1 | 0 | 0 | 1 (0.93) | | x | | | x | | | | *Tena et al. (2007), Bentur, Dalzell & Riordan (2007)* |
| *Corynebacterium casei* | GPB | 1 | 0 | 0 | 1 (0.93) | | | | | | | | x | *Bernard (2012)* |
| *Cronobacter sp.* | GNB | 0 | 1 | 0 | 1 (0.93) | x | x | | | x | | x | | *Patrick et al. (2014)* |
| *Delftia tsuruhatensis* | GNB | 2 | 0 | 0 | 2 (1.87) | x | | | | | | | | *Ranc et al. (2018)* |
| *Empedobacter falsenii* | GNB | 1 | 0 | 0 | 1 (0.93) | x | | x | | | | | | *Olowo-Okere et al. (2022)* |
| *Enterobacter kobei* | GNB | 0 | 0 | 1 | 1 (0.93) | x | | x | | x | x | | | *Hoffmann et al. (2005), Ji et al. (2021)* |
| *Enterobacter roggenkampii* | GNB | 0 | 0 | 1 | 1 (0.93) | | | x | | x | x | | | *Ji et al. (2021)* |
| *Enterococcus casseliflavus* | GPC | 0 | 0 | 1 | 1 (0.93) | | | | | x | | | | *Yoshino (2023)* |
| *Escherichia coli* | GNB | 1 | 0 | 0 | 1 (0.93) | | | x | x | x | x | | | *Mueller & Tainter (2023)* |
| *Lysinibacillus fusiformis* | GPB | 0 | 1 | 0 | 1 (0.93) | x | x | | | x | | | | *Sulaiman et al. (2018)* |
| *Macrococcus canis* | GPC | 1 | 1 | 0 | 2 (1.87) | | x | | | | | | | *Jost et al. (2021)* |
| *Macrococcus caseolyticus* | GPC | 7 | 1 | 1 | 9 (8.41) | | | | | | | x | | *Zhang et al. (2022)* |
| *Mammaliicoccus sciuri*[†] | GPC | 1 | 1 | 0 | 2 (1.87) | | x | x | | x | | | | *Boonchuay et al. (2023), Dakić et al. (2005)* |
| *Mixta calida* | GNB | 1 | 0 | 0 | 1 (0.93) | | | x | x | | | | | *Huzefa et al. (2022)* |
| *Moraxella osloensis* | GNCB | 0 | 0 | 2 | 2 (1.87) | | | x | x | | x | | | *Alkhatib et al. (2017)* |
| *Ochrombactrum amthropi* | GNB | 0 | 0 | 1 | 1 (0.93) | | | | | x | | | | *Jeyaraman et al. (2022)* |
| *Pantoea eucrina* | GNB | 0 | 1 | 0 | 1 (0.93) | | | | | | | x | | – |
| *Pantoea piersonii* | GNB | 1 | 0 | 0 | 1 (0.93) | | | | | | | x | | – |
| *Pluralibacter gergoviae* | GNB | 0 | 0 | 2 | 2 (1.87) | | x | x | | | | x | | *Furlan & Stehling (2023)* |
| *Priestia megaterium* | GPB | 0 | 0 | 1 | 1 (0.93) | | x | | x | x | | | | *Shwed et al. (2021), Bocchi et al. (2020)* |
| *Pseudomonas putida* | GNB | 0 | 2 | 0 | 2 (1.87) | x | x | | | x | | | | *Baykal et al. (2022)* |
| *Rothia amarae* | GPC | 0 | 1 | 1 | 2 (1.87) | | | | | | | x | | *Fatahi-Bafghi (2021)* |
| *Rothia endophytica* | GPC | 0 | 2 | 0 | 2 (1.87) | | | | | | | x | | *Fatahi-Bafghi (2021)* |

**Table 2** (*continued*)

| Microorganisms | Gram | No. of isolation | | | Total (%) (N = 107) | Infection systems | | | | | | | | Reference |
|---|---|---|---|---|---|---|---|---|---|---|---|---|---|---|
| | | Pork/Chicken (N = 52) | Seafood (N = 35) | Fish (N = 20) | | RT | SST | UT | CNS | BL | GI | Others | No report | |
| *Staphylococcus arlettae* | GPC | 1 | 0 | 0 | 1 (0.93) | | | | | x | | x | | *Dinakaran et al. (2012)* |
| *Staphylococcus aureus* | GPC | 0 | 0 | 1 | 1 (0.93) | x | x | | x | x | | x | | *Tong et al. (2015)* |
| *Staphylococcus epidermidis* | GPC | 3 | 0 | 0 | 3 (2.80) | | | | | x | | | | *Otto (2009)* |
| *Staphylococcus haemolyticus* | GPC | 0 | 0 | 1 | 1 (0.93) | | | | x | x | | x | | *Eltwisy et al. (2022)* |
| *Staphylococcus hominis* | GPC | 0 | 3 | 0 | 3 (2.80) | | | | | x | | x | | *Vasconcellos et al. (2022)* |
| *Staphylococcus kloosii* | GPC | 12 | 3 | 0 | 15 (14.02) | | x | | | x | | | | *Blondeau et al. (2021)* |
| *Staphylococcus saprophyticus* | GPC | 3 | 7 | 3 | 13 (12.15) | | | x | | | | x | | *Lawal et al. (2021)* |
| *Staphylococcus xylosus* | GPC | 1 | 1 | 0 | 2 (1.87) | | | | | | | | x | *Dordet-Frisoni et al. (2007)* |
| *Stenotrophomonas maltophilia* | GNB | 3 | 0 | 0 | 3 (2.80) | x | x | x | | x | | x | | *Brooke (2012)* |

**Notes.**

[†] It was previously known as *Staphylococcus sciuri*.

Abbreviations: N, number of isolates; RT, respiratory tract; SST, skin and soft tissues or wound; UT, urinary tract; CNS, central nervous system; BL, blood; GI, gastrointestinal tract; GNC, gram-negative cocci; GNCB, gram-negative coccobacilli; GNB, gram-negative bacilli; GPC, gram-positive cocci; GPB, gram-positive bacilli.

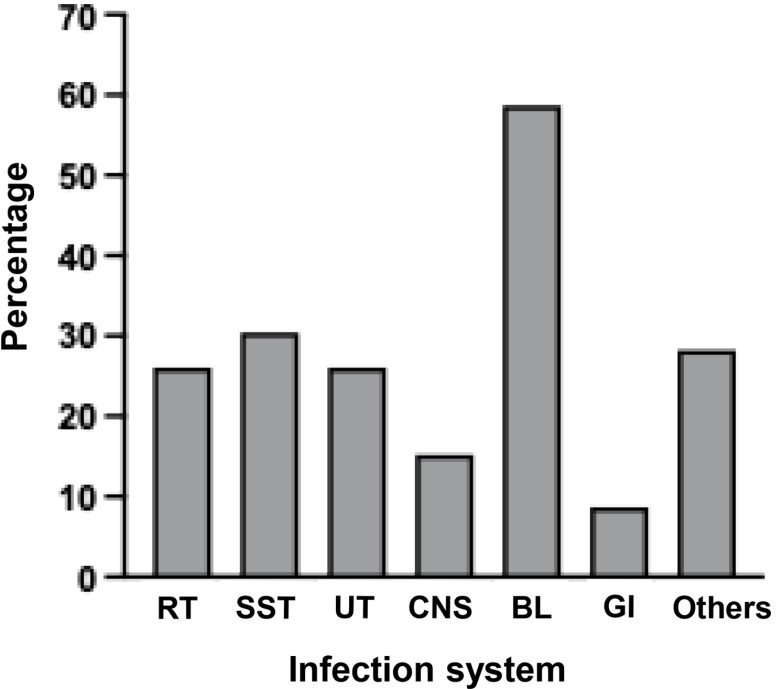

**Figure 4** Percentages of detected bacteria classified as human pathogens for causing infections in several systems.

For gram-negative bacteria, most of them were resistant to ampicillin, ampicillin/-clavulanic acid, and ceftazidime. Notably, *E. coli* and *P. putida* isolates were carbapenem-resistant, and *A. baumannii* isolates were MDR and carbapenem-resistant. Ceftazidime resistance was observed in particular *Acinetobacter* non-*baumannii* species, such as *A. bereziniae*, *A. defluvii*, *A. gandensis*, *A. seifertii*, and *A. ursingii*. Ciprofloxacin resistance was also discovered in *A. gandensis* (Table 3 and Table S5).

**Table 3  Antimicrobial resistance profiles of identified gram positive bacteria (A) and gram negative bacteria (B) in the study.**

| Microorganisms (N) | % Drug Susceptibility[a] | | | | | | | | | | | | | | | | | | |
|---|---|---|---|---|---|---|---|---|---|---|---|---|---|---|---|---|---|---|---|
| | P | DA | E | FOX | SXT | TE | FD | AMC | VA | h-CN | LEV | AMP | CAZ | CTX | CN | IMP | MEM | AK | CIP |
| **(A) Gram positive bacteria** | | | | | | | | | | | | | | | | | | | |
| *Staphylococcus* **(19)** | 21 | 74* | 89* | 95 | 100 | | | | | | | | | | | | | | |
| *S. kloosii* (4) | 0 | 50 | 100 | 100 | 100 | | | | | | | | | | | | | | |
| *S. hominis* (3) | 33 | 100 | 100 | 100 | 100 | | | | | | | | | | | | | | |
| *S. xylosus* (2) | 0 | 50 | 100 | 100 | 100 | | | | | | | | | | | | | | |
| *S. saprophyticus* (5) | 20 | 100 | 100 | 100 | 100 | | | | | | | | | | | | | | |
| *S. epidermidis* (2) | 100 | 100 | 100 | 100 | 100 | | | | | | | | | | | | | | |
| *S. arlettae* (1) | 0 | 0 | 0 | 100 | 100 | | | | | | | | | | | | | | |
| *S. haemolyticus* (1) | 0 | 0 | 0 | 0 | 100 | | | | | | | | | | | | | | |
| *S. aureus* (1) | 0 | 100 | 100 | 100 | 100 | | | | | | | | | | | | | | |
| *Macrococcus* **(6)** | 100 | 50 | 50 | 100 | 100 | 67* | 83* | | | | | | | | | | | | |
| *M. canis* (2) | 100 | 50 | 50 | 100 | 100 | 100 | 100 | | | | | | | | | | | | |
| *M. caseolyticus* (4) | 100 | 50 | 50 | 100 | 100 | 50 | 75* | | | | | | | | | | | | |
| *Enterococcus casseliflavus* (1) | 100 | | | | | | | 100 | 100 | 100 | | | | | | | | | |
| **(B) Gram negative bacteria** | | | | | | | | | | | | | | | | | | | |
| *Stenotrophomonas maltophilia* (1) | | | | | 100 | | | | | | 100 | | | | | | | | |
| *Enterobacter roggenkampii* (1) | | | | | 100 | | 0 | | | | | 0 | 0 | 100 | 100 | 100 | 100 | | |
| *Enterobacter kobei* (1) | | | | | 100 | | 0 | | | | | 0 | 0 | 0 | 100 | 100 | 100 | | |
| *Escherichia coli* (1) | | | | | 100 | | 0 | | | | | 0 | 0 | 100 | 100 | 0 | 100 | | |
| *Pseudomonas putida* (2) | | | | | | | | | | | | | 100 | | 100 | | 0 | 100 | 100 |
| *Acinetobacter* **(16)** | | | | | | | | | | | | | 50 | | 100 | | 88* | 94 | 88* |
| *A. baumannii* (2) | | | | | | | | | | | | | 0 | | 100 | | 0 | 50 | 50 |
| *A. bereziniae* (3) | | | | | | | | | | | | | 67* | | 100 | | 100 | 100 | 100 |
| *A. defluvii* (1) | | | | | | | | | | | | | 0 | | 100 | | 100 | 100 | 100 |
| *A. gandensis* (1) | | | | | | | | | | | | | 0 | | 100 | | 100 | 100 | 0 |
| *A. indicus* (1) | | | | | | | | | | | | | 100 | | 100 | | 100 | 100 | 100 |
| *A. junii* (1) | | | | | | | | | | | | | 100 | | 100 | | 100 | 100 | 100 |
| *A. pittii* (1) | | | | | | | | | | | | | 100 | | 100 | | 100 | 100 | 100 |
| *A. radioresistens* (1) | | | | | | | | | | | | | 100 | | 100 | | 100 | 100 | 100 |
| *A. seifertii* (1) | | | | | | | | | | | | | 0 | | 100 | | 100 | 100 | 100 |
| *A. ursingii* (3) | | | | | | | | | | | | | 33 | | 100 | | 100 | 100 | 100 |
| *A. variabilis* (1) | | | | | | | | | | | | | 100 | | 100 | | 100 | 100 | 100 |

**Notes.**

[a]The data was calculated from only susceptible result. Level of drug susceptibility was classified into three levels: Low susceptibility ($\leq 50\%$) was indicated in bold number; Intermediate susceptibility (51–89%) was indicated with asterisk sign (*); and High susceptibility ($\geq 90\%$).

Abbreviations: N, number of isolates; P, penicillin 10 µg; AMP, ampicillin 10 µg; AMC, ampicillin/clavulanic acid 20/10 µg; AK, amikacin 30 µg; DA, clindamycin 2 µg; E, erythromycin 15 µg; FOX, cefoxitin 30 µg; SXT, sulfamethoxazole/trimethoprim 23.75/1.25 µg; TE, Tetracyclin 30 µg; FD, Fusidic acid 10 µg; CN, gentamicin 10 µg; h-CN, gentamicin 120 µg; VA, vancomycin 30 µg; CAZ, ceftazidime 30 µg; CTX, cefotaxime 30 µg; MEM, meropenem 10 µg; IMP, imipenem 10 µg; CIP, ciprofloxacin 5 µg; LEV, levofloxacin 5 µg.

# DISCUSSION

Despite the digital world enabling cashless transactions, the utilization of currency remains essential for some individuals, locations, and circumstances. A significant quantity of germs has been found on banknotes, some of which are infectious to humans (*Phunpae et al.,*

*2018*; *Gosa, 2015*). The currency circulation could be a vehicle for transmitting pathogenic bacteria to other compromised or healthy individuals. Neglecting personal hygiene and allowing hand contact or physical transfer between bacteria-shedding sources and currency results in cash contamination. Sources of bacterial contamination in this study may occur *via* the release of infectious bacteria through mucus, feces, and aerosol droplets during coughing or sneezing by infected individuals, or from other contaminated environmental surfaces or fresh meats in each store. Paper notes from meat shops were likely to be contaminated with blood, which is a good medium for facilitating substantial microbial proliferation (*Yar, 2020*). Contaminated banknotes and coins act as a universal carrier for the spread of pathogens, posing a threat to public health.

Our finding aligned with prior studies (*Phunpae et al., 2018*; *Kalita et al., 2013*; *Glenn et al., 2015*) indicating that the positive culture results from banknotes were observed at a higher rate and quantity compared to coins. It is noteworthy that several bacterial strains were identified in the majority of positive culture samples, as indicated by previous research (*Basavarajappa, Rao & Suresh, 2005*). The difference in material type and size between Thai banknotes and Thai coins corroborated the findings. Nearly all Thai banknotes are composed of cotton-linen paper; however, the 20-baht banknote was changed to polymer material in 2009 to enhance durability for practical use. In contrast, Thai coins are composed of various metals based on their denomination, including copper, nickel, and aluminum (*Bank of Thailand, 2024*). The predominant metal in all Thai coins is copper, comprising between 75% and 92% (*Phunpae et al., 2018*). The porous structure of cotton-linen fibers in banknotes could promote bacterial colonization more effectively than polymers and metals (*Vriesekoop et al., 2016*). This may be attributed to a variety of physicochemical parameters in polymers (*Prasai, Yami & Joshi, 2010*) copper metal exhibits broad antibacterial properties that can suppress bacterial growth (*Salah, Parkin & Allan, 2021*).

Using the advanced MALDI-TOF MS technology to identify colonies grown on SBA/Mac agar facilitated this study to discover uncommon bacteria and newly emerging opportunistic pathogens that traditional biochemical assays, mostly utilized in prior research, could not identify. Our study identified a total of 107 bacterial isolates covering 46 species in 24 genera. Most isolates were identified as medically important and opportunistic pathogens, and the variation of bacterial species presented in currencies is comparable to previous studies (*Phunpae et al., 2018*; *Ofoedu et al., 2021*; *Yar, 2020*; *Ejaz, Javeed & Zubair, 2018*; *Gabriel, Coffey & O'Mahony, 2013*; *Alemu, 2014*). Despite the rarity of these uncommon bacteria infecting immunocompromised hosts, our comprehensively reviewed demonstrated that opportunistic and nosocomial infections have been recently documented.

Genus *Staphylococcus* was the most contaminated bacteria on cash samples that this study could identify. The eight species included one coagulase-positive *S. aureus* and seven coagulase-negative staphylococci (CoNS), with a distinction in frequency. Staphylococci are common skin and mucous membrane colonizers in humans, poultry, and other warm-blooded animals (*Lee & Yang, 2021*). Despite coagulase-negative staphylococci (CoNS) being less prevalent in human pathogenesis compared to *S. aureus*, its potential to cause infections in humans and animals under appropriate conditions has drawn greater scientific

interest over the past decade (*Von Eiff, Peters & Heilmann, 2002*). Numerous species of coagulase-negative staphylococci, including *S. gallinarum*, *S. arlettae*, *S. chromogenes*, *S. xylosus*, and *S. epidermidis*, have been frequently isolated from the nares and skin of chickens. Our investigation was able to identify *S. arlettae*, *S. xylosus*, and *S. epidermidis* from pork and chicken stores. Recent studies have demonstrated that CoNS can also produce staphylococcal enterotoxins (SEs) and could be a potential cause of food poisoning. Numerous virulence genes linked to pathogenesis, such as *ica*, *nuc*, and *ssp*, which are typically present in the genomes of pathogenic staphylococci, are also identified in specific CoNS, including *S. haemolyticus*, *S. saprophyticus*, and *S. arlettae* (*Shimizu et al., 1992*; *Lavecchia et al., 2019*).

Furthermore, the potential role of CoNS in the transmission of antimicrobial resistance by serving as a reservoir for antimicrobial resistance genes has been increasingly documented (*Nemeghaire et al., 2014*; *Archer & Niemeyer, 1994*). Staphylococci that were nearly entirely isolated, with the exception of *S. epidermidis*, demonstrated a high level of penicillin resistance. From the late 1960s to the present, over 80% of staphylococcal isolates acquired in both community and hospital settings were resistant to penicillin, a resistance that was mediated by the *blaZ* gene, which encodes $\beta$-lactamase/penicillinase (*Lowy, 2003*). Only *S. haemolyticus* exhibited a methicillin-resistant (MR) strain, while *S. haemolyticus* and *S. arlettae* were multidrug-resistant strains (resistant to three or more antibiotic classes). This study was in line with the previous study, which could isolate MDR-*S. heamolyticus* from ready-to-eat (RTE) foods served in bars and restaurants (*Chajecka-Wierzchowska et al., 2023*). Despite the legal prohibitions, antimicrobial agents are still in use for prophylaxis or metaphylaxis in aquaculture and livestock production (*Collignon & McEwen, 2019*; *Pepi & Focardi, 2021*). In recent years, there has been a significant increase in the number of reports on the occurrence of antibiotic-resistant CoNS in food (*Aslantaş & Yıldırım, 2021*; *Silva et al., 2022*). This strongly implies that the food chain production may represent a pathway for the transmission of antimicrobial resistance genes. Antimicrobial resistance genes in staphylococci are typically located on plasmids, transposons, or other mobile genetic elements (MGEs), facilitating horizontal gene transfer (*El-Adawy et al., 2022*).

*Acinetobacter* was the second most prevalent bacterium on currency samples; twelve species of *A. baumannii* and eleven *Acinetobacter* non-*baumannii* (*Anb*) were detected. *Anb* species are becoming more significant as opportunistic nosocomial pathogens for humans (*Sheck et al., 2023*; *Wong et al., 2017*). The identified strains of *A. lactucae* (formerly referred to as *A. dijkshoorniae*), *A. pittii*, and *A. seifertii* belong to the *Acinetobacter calcoaceticus-baumannii* (*Acb*) complex (*Sheck et al., 2023*). While the phenotypes of the species within the *Acb* complex are nearly identical, they exhibit substantial differences in terms of their ecology, pathogenesis, epidemiology, and susceptibility to antibiotics (*Nemec et al., 2011*). Moreover, the isolated isolates of *A. bereziniae*, *A. junii*, and *A. ursingii* have been frequently reported in human infections (*Sheck et al., 2023*).

The presence of extensive antibiotic resistance phenotype in *A. baumannii* isolated from cash samples in this study, as well as from commercial food samples in prior research (*Al Atrouni, 2016*; *Lupo, Haenni & Madec, 2018*), indicated that environmental sources may contribute to the transmission of antibiotic-resistant bacteria to humans. *A. baumannii*

exhibited intrinsic resistance to numerous antibiotics and possessed a notable capacity to acquire resistance to all existing therapeutic agents, including carbapenems. This has established it among the ESKAPE pathogens (*E. faecium*, *S. aureus*, *K. pneumoniae*, *A. baumannii*, *P. aeruginosa*, and *Enterobacter* spp.) (*Boucher et al., 2009*) and contributed to the critical position on the WHO's global priority list of antibiotic-resistant bacteria (*Breijyeh, Jubeh & Karaman, 2020*). Nonetheless, most of the isolated *Anb* demonstrated susceptibility to the majority of the tested antibiotics, with the exception of certain species that exhibited resistance to ceftazidime. This aligns with earlier research that identified ceftazidime resistance in *Anb* isolates at a rate of 12.6% (*Kittinger et al., 2017*). Moreover, nearly all other gram-negative bacteria exhibited resistance to penicillins and third-generation cephalosporins. Resistance of *Enterobacteriaceae* to third-generation cephalosporins exceeds 10%, whereas resistance to carbapenems ranges from 2% to 7%. This is a result of the rapid dissemination of strains that produce extended-spectrum β-lactamases (ESBLs) (*Baba et al., 2009*).

*Macrococcus* was another type of isolate that was prevalent in this investigation. The genus *Macrococcus* is closely related to the genus *Staphylococcus* (*Baba et al., 2009*; *Mazhar et al., 2019*). It demonstrated significant homology in phenotypic and biological traits, sharing characteristics with these oxidase-positive and novobiocin-resistant staphylococci (*Mašlaňová et al., 2018*). Members of *Macrococcus* are currently classified into twelve species (*Mašlaňová et al., 2018*), which are commonly isolated from animal skin (ponies, horses, cows, llamas, and dogs), as well as from dairy or meat products (*Kloos et al., 1998*; *Mannerová et al., 2003*; *Gobeli Brawand et al., 2017*). Despite the almost all macrococci have yet to be documented in human clinical specimens, recent studies indicated high mortality rates in animals infected with Macrococci (*Cotting et al., 2017*; *Gobeli Brawand et al., 2017*; *Li et al., 2018*), suggesting a potential risk for human infections. *Macrococcus caseolyticus* and *M. canis* were two isolated species in the present study. *M. caseolyticus* was the most prevalent species that was previously classified as *Staphylococcus caseolyticus* due to the high degree of similarity (*Cotting et al., 2017*). Recently, *M. canis* was described for the first time as isolated from a human skin infection (*Jost et al., 2021*).

It is crucial to note that the adaptive acquisition of methicillin resistance genes in Macrococci genomes, such as *mecABCD* has been observed over the past decades (*Jost et al., 2021*; *Schwendener, Cotting & Perreten, 2017*; *Gómez-Sanz et al., 2015*). In contrast to classical *mecA* and *mecC*, which are predominantly associated with *S. aureus*, the methicillin-resistant genes found in Macrococci are primarily associated with *mecB* and *mecD* (*Zhang et al., 2022*; *Tsubakishita et al., 2010*). The *mecB* and *mecD* genes, being homologs of *mecA*, raise significant safety concerns because of the potential for these mobile elements to transfer to other commensal or pathogenic bacteria, such as *S. aureus* (*Chanchaithong, Perreten & Schwendener, 2019*; *Gómez-Sanz et al., 2015*). Despite the fact that this investigation was unable to identify methicillin-resistant *Macrococcus* spp. in Thai currency samples, the isolation of a potential multidrug-resistant strain may have been the result of the acquisition of MDR mobile genetic elements (*Zhang et al., 2022*).

## CONCLUSIONS

Despite the advanced features of E-commerce technology, the use of currency as a universal medium for daily transactions remains indispensable. This study demonstrated that Thai currency, particularly Thai banknotes, was significantly contaminated with numerous highly pathogenic and newly emerging opportunistic microorganisms. It is crucial to note that the majority of these bacteria were resistant to antibiotic therapeutic medications, which raises concerns about the public health risks associated with the potential transmission of pathogens through contaminated cash. Furthermore, certain bacteria have significant drug-resistant genes within their genome, posing a risk of transferring these mobile genetic elements to other commensal or pathogenic bacteria. Consequently, our study indicates that increased focus is necessary on the surveillance and monitoring of bacterial contamination on currency, while also promoting the use of digital wallets to reduce exposure to contaminated materials, alongside enhancing education on maintaining proper personal hygiene practices.

### Funding

This work was supported by the Thailand Science Research and Innovation (TSRI) Fundamental Fund, fiscal year 2024. The funders had no role in study design, data collection and analysis, decision to publish, or preparation of the manuscript.

### Grant Disclosures

The following grant information was disclosed by the authors:
The Thailand Science Research and Innovation (TSRI) Fundamental Fund, fiscal year 2024.

### Competing Interests

The authors declare there are no competing interests.

### Author Contributions

- Nattamon Niyomdecha conceived and designed the experiments, performed the experiments, analyzed the data, prepared figures and/or tables, authored or reviewed drafts of the article, and approved the final draft.
- Suwitchaya Sungvaraporn performed the experiments, analyzed the data, prepared figures and/or tables, authored or reviewed drafts of the article, and approved the final draft.
- Arisa Pinmuang performed the experiments, analyzed the data, prepared figures and/or tables, authored or reviewed drafts of the article, and approved the final draft.
- Narissara Mungkornkaew performed the experiments, analyzed the data, prepared figures and/or tables, and approved the final draft.
- Thanchira Saita performed the experiments, analyzed the data, prepared figures and/or tables, and approved the final draft.

- Waratchaya Rodraksa performed the experiments, analyzed the data, prepared figures and/or tables, and approved the final draft.
- Achiraya Phanitmas performed the experiments, analyzed the data, prepared figures and/or tables, and approved the final draft.
- Nattapong Yamasamit performed the experiments, analyzed the data, prepared figures and/or tables, authored or reviewed drafts of the article, and approved the final draft.
- Pirom Noisumdaeng conceived and designed the experiments, performed the experiments, analyzed the data, prepared figures and/or tables, authored or reviewed drafts of the article, project administration and Funding acquisition, and approved the final draft.

## Data Availability

The raw data are available in the Supplementary Files.

## Supplemental Information

Supplemental information for this article can be found online at http://dx.doi.org/10.7717/peerj.19465#supplemental-information.

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
