# Peer review of "Identification of bacteria on Thai banknotes and coins using MALDI-TOF mass spectrometry and their phenotypic antimicrobial susceptibility profiles"

_PeerJ, doi:10.7717/peerj.19465_

## Round 0.1 · original submission · Major Revisions

Please address the reviewers' comments.

Reviewer 1 ·

Basic reporting

In this study, the authors investigated bacterial contaminants on banknotes and coins using MALDI-TOF Mass Spectrometry and determined antimicrobial susceptibility testing (AST) in selected isolates. While the manuscript presents interesting data, several areas require clarification and improvement before further consideration.
1. The authors should specify the criteria for selecting the 150 banknotes and coins used in the study.
2. Additional samples from other sources, such as vegetable and grocery stores, should be included to provide more comprehensive insights into the prevalence of medically important bacterial pathogens.
3. I recommend using phosphate-buffered saline (PBS) instead of TSB to resuspend the swab.
4. Bacterial identification should be performed directly from the plates used for colony counting to enhance the accuracy of results.
5. The authors reported a higher contamination rate on banknotes than on coins, possibly due to differences in size or surface area. However, the comparison between banknotes and coins may not be justified. Instead, a comparison of contamination levels across different collection sites for both banknotes and coins would be more informative.
6. The study focuses solely on bacterial contamination. The authors should also examine fungal pathogens to provide a more comprehensive analysis.
7. Based on the results, the variation among bacterial contaminants appears minimal. Further clarification is needed to support this conclusion.
8. References should be included throughout the methodology section, and detailed information about the database used for bacterial identification should be provided.

Experimental design

The experimental design requires improvement.

Validity of the findings

Validity of the findings looks good to me

Additional comments

NA

Reviewer 2 ·

Basic reporting

The paper is well-designed and written. However, there are some typos and suggestions were made.

Experimental design

Well designated, but needs referencing, see the attached file

Validity of the findings

Acceptable

Additional comments

Please find the comments on the attached file

Annotated reviews are not available for download in order to protect the identity of reviewers who chose to remain anonymous.

---

## Round 0.2 · accepted · Accept

Thanks for addressing the reviewers' comments!

Reviewer 1 ·

Basic reporting

no comment

Experimental design

no comment

Validity of the findings

no comment

Additional comments

The current version of the manuscript has improved significantly and is now well-structured and comprehensive. The authors have adequately addressed all the queries and concerns I previously raised and have provided sufficient and necessary information throughout the revised manuscript.

I have only one remaining suggestion: please include appropriate references for the Antimicrobial Susceptibility Testing (AST) methodology in the Methods section to enhance clarity and reproducibility.

Overall, I commend the authors for their thorough revisions and believe the manuscript is now suitable for publication pending the minor suggestion above.

Reviewer 2 ·

Basic reporting

Thank you. I have checked the edited version of the manuscript and compared it with the first draft. It seems that the author has implemented most of the suggestions and corrections. I think the paper is now acceptable for publication.

Experimental design

The manuscript is now clear and well-organized. I have reviewed the edited version and compared it with the original draft. The author has addressed most of the suggested revisions

Validity of the findings

Acceptable

Additional comments

Thank you